# Shifts in the epidemic season of human respiratory syncytial virus associated with inbound overseas travelers and meteorological conditions in Japan, 2014–2017: An ecological study

Keita Wagatsuma[1]*, Iain S. Koolhof[2], Yugo Shobugawa[3], Reiko Saito[1,4]

1 Division of International Health (Public Health), Graduate School of Medical and Dental Sciences, Niigata University, Niigata, Japan, 2 College of Health and Medicine, School of Medicine, University of Tasmania, Hobart, Tasmania, Australia, 3 Department of Active Aging, Graduate School of Medical and Dental Sciences, Niigata University, Niigata, Japan, 4 Infectious Diseases Research Center of Niigata University in Myanmar (IDRC), Graduate School of Medical and Dental Sciences, Niigata University, Niigata, Japan

* waga@med.niigata-u.ac.jp

**Data Availability Statement:** All relevant data are within the paper and its Supporting Information files.

## Abstract

Few studies have examined the effects of inbound overseas travelers and meteorological conditions on the shift in human respiratory syncytial virus (HRSV) season in Japan. This study aims to test whether the number of inbound overseas travelers and meteorological conditions are associated with the onset week of HRSV epidemic season. The estimation of onset week for 46 prefectures (except for Okinawa prefecture) in Japan for 4-year period (2014–2017) was obtained from previous papers based on the national surveillance data. We obtained data on the yearly number of inbound overseas travelers and meteorological (yearly mean temperature and relative humidity) conditions from Japan National Tourism Organization (JNTO) and Japan Meteorological Agency (JMA), respectively. Multi-level mixed-effects linear regression analysis showed that every 1 person (per 100,000 population) increase in number of overall inbound overseas travelers led to an earlier onset week of HRSV epidemic season in the year by 0.02 week (coefficient –0.02; P<0.01). Higher mean temperature and higher relative humidity were also found to contribute to an earlier onset week by 0.30 week (coefficient –0.30; P<0.05) and 0.18 week (coefficient –0.18; P<0.01), respectively. Additionally, models that included the number of travelers from individual countries (Taiwan, South Korea, and China) except Australia showed that both the number of travelers from each country and meteorological conditions contributed to an earlier onset week. Our analysis showed the earlier onset week of HRSV epidemic season in Japan is associated with increased number of inbound overseas travelers, higher mean temperature, and relative humidity. The impact of international travelers on seasonality of HRSV can be further extended to investigations on the changes of various respiratory infectious diseases especially after the coronavirus disease 2019 (COVID-19) pandemic.

**Funding:** This study was supported by the Japan Initiative for Global Research Network on Infectious Diseases (J-GRID) from the Japan Agency for Medical Research and Development (AMED) [grant number 15fm0108009h0001-19fm0108009h0005]; and the Health and Labor Sciences Research Grants, Ministry of Health, Labor and Welfare, Japan [grant number H30-Shinkougyousei-Shitei-002] (to RS). The funders had no role in study design, data collection and analysis, decision to publish, or preparation of the manuscript.

**Competing interests:** The authors have declared that no competing interests exist.

## Introduction

Human respiratory syncytial virus (HRSV) is an infection of the respiratory tract which causes clinically severe pneumonia in children and bronchiolitis in infants [1]. Globally, acute lower respiratory tract infections caused by HRSV lead to approximately 70,000 annual deaths in children under the age of 5 years and approximately 3.4 million people require hospitalization [2]. Epidemics of HRSV are seasonal, typically having an epidemic during winter in temperate zones as seen in Japan [3]. The HRSV epidemic season in Japan normally begins around October, peaks in December-January, and ends in March-April. However, a shift in the HRSV epidemic season to summer periods (from June-August) has been reported in Japan more recently [4]. Although many previous studies looked at the association between HRSV cases and meteorological conditions [3,5–11], there may well be other factors that determine the onset week of HRSV epidemic season.

In recent years, with a large number of travelers coming into Japan, the likelihood for contact the local population have with foreign travelers has increased [12]. Until the start of coronavirus disease 2019 (COVID-19) pandemic in 2020, the number of inbound overseas travelers increased in the last several decades and was expected to continue to increase in the future. Several studies have suggested the possibility of increased transmission of infectious diseases owing to greater international travel and arrivals [13–16], suggesting that the inflow of infected travelers may be associated with the epidemics. Indeed, we recently reported that early cases of COVID-19 in January-February 2020 in Japan were concentrated in prefectures where the numbers of foreign travelers were high [17].

This study aims to test whether the number of inbound overseas travelers and meteorological conditions are associated with the onset week of HRSV epidemic season in Japan. We included the number of inbound overseas travelers as a parameter as well as meteorological conditions, to examine the associations with the onset week of HRSV epidemic season during 2014–2017.

## Materials and methods

### Study design

This study was designed to evaluate the ecological association between meteorological (mean temperature and relative humidity) conditions and the number of inbound travelers on the onset week of HRSV epidemic season in Japan over a 4-year period (2014–2017).

### Study area

The study area is Japan, which consists of 47 prefectures. Of the 47 prefectures, 46 prefectures, excluding Okinawa prefecture, belong to the temperate zones [3]. Okinawa belongs to the subtropical region where the seasonality of HRSV is different [18]. Therefore, data for this study, only included the 46 prefectures which have temperate climates and excluded the subtropical Okinawa prefecture.

### HRSV national surveillance data in Japan

HRSV epidemiological data, available from the Infectious Disease Weekly Report (IDWR), sourced from the National Epidemiological Surveillance of Infectious Diseases (NESID), published by National Institute of Infectious Diseases (NIID) under the Ministry of Health, Welfare and Labour in Japan (MHLW) was used for the analysis in this paper [19]. The MHLW designates approximately 3,000 paediatric sentinel sites (i.e., hospitals and clinics) in Japan, which report numbers of patients diagnosed as HRSV infection at weekly basis to the prefecture or municipal public health sectors in Japan [18]. An HRSV case is defined by a positive

rapid diagnostic test (RDT) kits licenced in Japan, or a laboratory confirmation such as virus isolation or antibody rise in paired sera according to the MHLW guidelines [20]. The prefectural data is reported to the NIID, and the collective numbers of HRSV cases per prefecture are released weekly through its website [19]. However, the information at the individual paediatric sentinel sites where each HRSV case count is reported is anonymised and not made public. Therefore, we could not specify whether these NIID data came from the same sentinel sites during the study period. Still we assume these changes of sentinel sites were minor and did not affect the quality of data. A sentinel site is located roughly over 30,000 populations according to the Infectious Disease Control Law in Japan and the prefecture government are responsible for maintaining the sites so that it does not affect the quality of data [21].

In this study, we extracted the total number of HRSV cases in each prefecture (excluding Okinawa prefecture) reported during the 1st week of 2014 and the 52nd week of 2018 from the NESID data by weekly basis. Since the number of sentinel sites differed by prefecture largely due to population size in the area, then the numbers of HRSV cases per sentinel site (numbers of patient visits due to HRSV cases per hospital/clinic) was calculated from the total number of cases divided by the number of sentinel sites. Using the number of weekly HRSV cases per sentinel site, we created the epidemic curve of HRSV in Japan from 2014 to 2017.

## Definition of onset week of HRSV epidemic season

We used the onset week of HRSV epidemic season in 46 prefectures for 4-year period estimated by Yamagami et al. in the other paper [4]. They developed a new algorithm to estimate the start of the HRSV epidemic season based on the IDWR data, since NIID does not set the definition on the start of HRSV epidemic season in Japan. In their paper, in determining the epidemic starting point of the HRSV epidemic season in each prefecture, they make estimates that take into account both the number of HRSV reports included within the epidemic period relative to the total number of HRSV reports (capture rate) and the number of HRSV reports per week within the epidemic period (HRSV-reports/w). They calculate both these capture rates and the HRSV-reports/w search index to determine the starting point of the epidemic cycle (see below Eqs (1–3)).

In the following, Yamagami et al. [4] define the parameters for A(x) denotes HRSV-reports/w (the number of HRSV reported per week); B(x) denotes capture rate (the number of HRSV reports included within onset-trough periods relative to the total number of HRSV reports in the dataset); Index(x) denotes the search index. The capture rate A(x) and HRSV-reports/w (B(x)) were converted into the same scale and summed; x is a percentile value ranging from 0 to 100 in integer units; s is the number of epidemic cycles in the dataset; Onset(x) is the ordinal number of each onset week within the dataset from 1 to x in integer units; Trough is the ordinal number of each trough-week within the dataset from 0 to x in integer units; k is the number of HRSV reports in a given week.

$$A(x) = \frac{\sum_{i=0}^{s} \sum_{j=Onset(x)_i}^{Trough_i} k_j}{\sum_{i=1}^{s} (Trough_i - Onset(x)_i + 1)} \tag{1}$$

$$B(x) = \frac{\sum_{i=1}^{s} \sum_{j=Onset(x)_i}^{Trough_i} k_j}{\sum_{j=Trough(0)}^{Trough(s)} k_j} \tag{2}$$

$$Index(x) = \frac{A(x)}{Maximum\{x|A(x)\}} + \frac{B(x)}{Maximum\{x|B(x)\}} \tag{3}$$

The procedure for setting up the onset week of HRSV epidemic season by Yamagami et al. [4] was defined as follows. First, they transposed the number of HRSV reports in each prefecture to the percentile rank and selected the lowest percentile rank that was greater than the percentile value. Secondly, they selected the 101st percentile rank in each prefecture for the 101st percentile value. Third, they defined the reporting week corresponding to the intersection of these percentile ranks and the increase in the slope of the sine curve as the onset week. Finally, they applied this generation procedure to all 47 prefectures in Japan to estimate the onset week at prefectural level. The validity of the determined onset was confirmed by calculating the capture rate. The validity of the model was subsequently tested using data from 47 prefectures in the previous study [4].

In this study, the onset weeks of HRSV epidemic season by prefecture over the 4-year period from 2014 to 2017 were extracted from their paper [4] and used in the analysis.

## Dependent variable

The dependent variable for the models is the onset week of HRSV epidemic season, ranged from weeks 1–52, by each prefecture [4].

## Explanatory variables

**Meteorological data.** The meteorological data published by the Japan Meteorological Agency (JMA) [22] was used as explanatory variables for the models in this paper. Monthly and yearly meteorological data observed at a meteorological observatory situated at the prefectural capital city was used for each respective prefecture. We extracted monthly and yearly mean temperature and relative humidity from 2014 to 2017. There were two meteorological observatories with missing data on relative humidity. Therefore, we selected the observatories sites with the nearest distance from the prefectural capital by substitution. Using these monthly meteorological data in each prefecture, mean temperature and relative humidity over the whole of Japan have been calculated.

**Inbound overseas travelers.** As a surrogate of the monthly and yearly number of overall inbound overseas travelers by prefecture, the daily number of foreign visitors accommodated (e.g., hotels) in each prefecture from 2014 to 2017 were sourced from the travel statistics survey of the Japan National Tourism Organization (JNTO) [23]. In addition, we used yearly number of travelers by country from the top three countries (China, South Korea, and Taiwan) with the most travelers coming into Japan. In addition, Australia was used as the country with the largest number of travelers from the southern hemisphere from 2014 to 2017.

## Ethical considerations

This ecological study meets the ethical and regulatory guidelines, including adherence to the legal requirements of the study country. The collection of HRSV data under the NESID is legally defined and implemented under the Infectious Disease Control Law of the Japanese government. Through this scheme, the requirement for informed consents from the patients were legally waived. The MHLW publishes these data as a web-based open source data [19]. Similarly, the meteorological data and the inbound overseas traveler's data are also open to the public from the respective governmental website [22,23].

## Statistical analysis

The statistical analyses were conducted in two steps. Firstly, before the main analysis, the association between each variable included in this study, onset week of HRSV epidemic season,

mean temperature, relative humidity, number of overall inbound overseas travelers per 100,000 population by each prefecture in Japan, number of inbound overseas travelers from Taiwan, South Korea, China, and Australia per 100,000 populations by each prefecture in Japan, was calculated using Spearman's rank test to evaluate the correlation between variables (N = 4×46 = 184). Subsequently, variables that showed strong multicollinearity were excluded in the regression model.

Secondly, multi-level mixed-effects linear regression model was performed to adjust for the effects of confounding variables (N = 4×46 = 184). We conceptualised our analysis with a multi-level structure consisting of factors nested within prefectures (prefectural-level) to account for the impact of year repeated measures. We fitted the data using a multi-level linear regression procedure with a mixed effect model, adjusting for both individual and prefectural levels as fixed effects. The parameters of the model were chosen to be random effects estimators using the maximum likelihood. The onset week of HRSV epidemic season by each year (continuous) was the dependent variable, and yearly number of inbound travelers (continuous) and meteorological (yearly mean temperature and relative humidity) conditions (continuous) were included as explanatory variables. Furthermore, the model was adjusted by inputting year variables (years 2014, 2015, 2016, and 2017) (dummy variables) as a covariate to control seasonal variation. Our first analysis involved the estimation of a null model (Model 1). The null model allows us to decompose the variance in the onset week of HRSV epidemic season to determine whether it is due to variation at the prefectural or individual level. Model 2 represents the number of overall inbound travelers per 100,000 population. Models 3, 4, 5, and 6 consider the numbers of inbound travelers per 100,000 population from Taiwan, South Korea, China, and Australia, respectively. The regression coefficients and P-values were calculated in each multi-level analysis. Goodness-of-fit statistics of each model was evaluated by calculating the Akaike's Information Criterion values (AIC). The intra-class correlation (ICC) was calculated to evaluate the similarity within group.

All tests were two-sided and P-values less than 0.05 were considered statistically significant. All data analyses were performed using STATA software, Ver. 15.0 (Stata Corp LP College Station, TX, USA).

## Results

### Seasonality trends of HRSV cases, mean temperature, relative humidity, and inbound overseas travelers throughout Japan, 2014–2017

HRSV activity in Japan usually peaks in the winter season (from December-January). The peak epidemic of HRSV was observed in December in 2014 and 2015, but was changed to October in 2016, and to September in 2017 (Fig 1A, S1 Dataset). Correspondingly, the annual onset week of the HRSV epidemic season occurred earlier in the later year. The median onset weeks of the HRSV epidemic season were around 36.5 weeks (September) in 2014, 36.0 weeks (September) in 2015, 35 weeks (late August) in 2016, and 30 weeks (beginning of July) in 2017, respectively. The median onset week of HRSV epidemic season was approximately 6.0 weeks (1.5 months) earlier in 2017 than in 2014. On the other hand, the number of inbound overseas travelers increased year on year, doubled from 13,413,467 in 2014 to 28,691,073 in 2017 (S2 Dataset). However, the monthly number of inbound overseas travelers from all countries during 2014 and 2017 did not show clear seasonal trend (Fig 1B and S2 Dataset). The number of inbound overseas travelers from Taiwan, China and South Korea remained almost steady through the year, while Australia is observed to have a higher number of inbound travelers in the winter months of January and December (Fig 1C–1F, and S3–S6 Datasets). This pattern did not differ during the four years. The mean temperature peaks at approximately 27˚C in

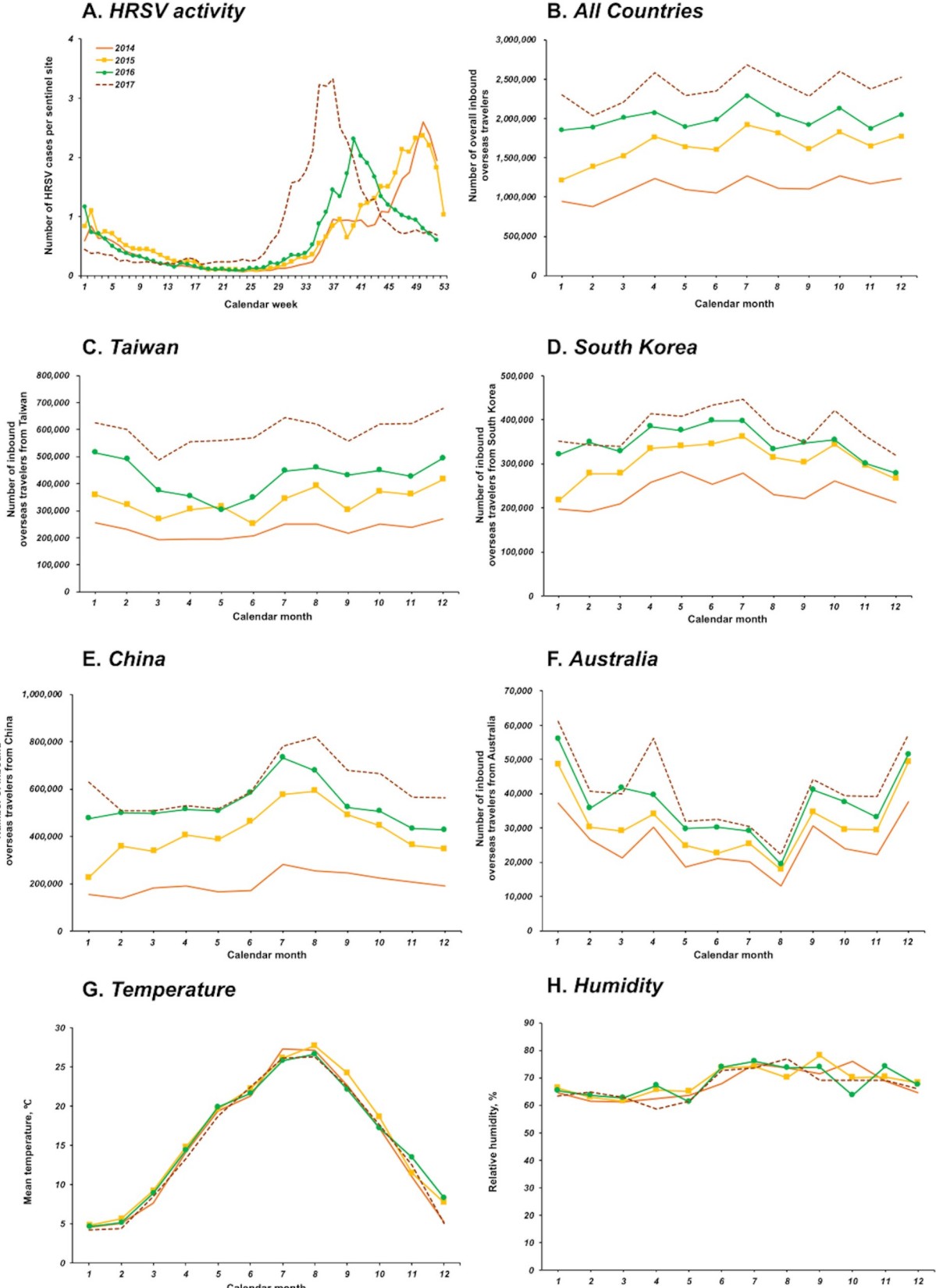

**Fig 1. Trends of HRSV cases, mean temperature, relative humidity, and inbound overseas travelers throughout Japan, 2014–2017.** (A) Epidemic curve of HRSV in Japan from 2014 to 2017. (B) Monthly number of overall inbound overseas travelers from all countries during 2014 and 2017, (C) Taiwan, (D) South Korea, (E) China, (F) Australia, (G) Monthly mean temperature (˚C) throughout Japan from 2014 to 2017, (H) Monthly average relative humidity (%) from 2014 to 2017. The orange, yellow, green and brown lines represent 2014, 2015, 2016 and 2017, respectively.

August of each year and the relative humidity is around 60–70% throughout the year in Japan (Fig 1G and 1H, S7 and S8 Datasets). Mean temperature and relative humidity did not vary significantly throughout the 4-year period.

## Spearman's correlation analysis of variables included in the study

Table 1 shows the matrix of Spearman's correlation coefficients [ρs] for variables included in the study for 4-year period (S9 Dataset). Onset week of HRSV epidemic season and mean temperature showed significant inverse correlation (Spearman's ρ = –0.17). Onset week of HRSV epidemic season was also significant having an inverse correlation with the number of overall inbound travelers per 100,000 population and number of inbound travelers from individual country per 100,000 population (Spearman's ρ = –0.20, –0.16, –0.32, –0.16 and –0.15, respectively). These results suggested the increased temperature and the higher number of overseas travelers are associated to earlier onset of HRSV season, respectively.

## Association between meteorological conditions and number of inbound travelers on onset week of HRSV epidemic season in Japan by multi-level analysis

The multi-level analysis showed that mean temperature, relative humidity, and the number of inbound travelers were negatively associated with the onset week of the HRSV epidemic season (Table 2, S9 Dataset). The null model with no predictors (Model 1) revealed no significant evidence of inter-prefectural variation (prefectural-level variance = 0.61, ICC = 0.01). In Model 2, which included the number of overall inbound travelers, we found that every 1 degree increases in mean temperature led to an earlier onset week within the year by 0.30 week (Model 2: coefficient –0.30; P<0.05). In addition, every 1.0% increase in relative humidity and every 1 person (per 100,000 populations) increase in the number of overall inbound travelers were associated with an earlier onset week of the HRSV epidemic season by 0.18 week and

**Table 1. Spearman's correlation of variables included in the study (N = 184).**

| Variable | 1 | 2 | 3 | 4 | 5 | 6 | 7 | 8 |
|---|---|---|---|---|---|---|---|---|
| 1. Onset week of HRSV epidemic season, (week) | 1.00 | | | | | | | |
| 2. Mean temperature, (˚C) | –0.17[a] | 1.00 | | | | | | |
| 3. Relative humidity, (%) | –0.11 | –0.12 | 1.00 | | | | | |
| 4. Overall inbound travelers per 100,000 population, (person) | –0.20[b] | –0.24[c] | –0.49[c] | 1.00 | | | | |
| 5. Inbound travelers from Taiwan per 100,000 population, (person) | –0.16[a] | 0.15[a] | –0.43[c] | –0.93[c] | 1.00 | | | |
| 6. Inbound travelers from South Korea per 100,000 population, (person) | –0.32[c] | 0.39[c] | –0.18[a] | 0.79[c] | 0.72[c] | 1.00 | | |
| 7. Inbound travelers from China per 100,000 population, (person) | –0.16[a] | 0.20[b] | –0.51[c] | 0.94[c] | 0.82[c] | 0.69[c] | 1.00 | |
| 8. Inbound travelers from Australia per 100,000 population, (person) | –0.15[a] | –0.05 | –0.51[c] | 0.86[c] | 0.84[c] | 0.53[c] | 0.82[c] | 1.00 |

HRSV, human respiratory syncytial virus.

[a] P<0.05.

[b] P<0.01.

[c] P<0.001.

**Table 2. Regression coefficients with P-values for onset week of HRSV epidemic season: The results of multi-level mixed-effects linear regression analysis (N = 184).**

| Variable | Model 1 | Model 2 | Model 3 | Model 4 | Model 5 | Model 6 |
|---|---|---|---|---|---|---|
| | Coef. | Coef. | Coef. | Coef. | Coef. | Coef. |
| Mean temperature, (˚C) | | –0.30[a] | –0.34[b] | –0.29[a] | –0.30[a] | –0.30[a] |
| Relative humidity, (%) | | –0.18[b] | –0.20[b] | –0.17[b] | –0.18[b] | –0.17[a] |
| Overall inbound travelers per 100,000 population, (person) | | –0.02[b] | | | | |
| Inbound travelers from Taiwan per 100,000 population, (person) | | | –0.07[b] | | | |
| Inbound travelers from South Korea per 100,000 population, (person) | | | | –0.25[b] | | |
| Inbound travelers from China per 100,000 population, (person) | | | | | –0.09[a] | |
| Inbound travelers from Australia per 100,000 population, (person) | | | | | | –0.60 |
| Year | | | | | | |
| 2014 | | Ref. | Ref. | Ref. | Ref. | Ref. |
| 2015 | | –0.86 | –0.81 | –0.86 | –0.80 | –0.94 |
| 2016 | | –2.59[c] | –2.54[c] | –2.58[c] | –2.55[c] | –2.69[c] |
| 2017 | | –7.77[c] | –7.79[c] | –7.60[c] | –7.76[c] | –7.87[c] |
| Intercept | 34.23[c] | 55.41[c] | 57.06[c] | 54.49[c] | 55.07[c] | 54.24[c] |
| Mixed effect | | | | | | |
| Prefectural-level variance (SE) | 0.61 (1.00) | 1.37 (0.29) | 1.30 (0.29) | 1.26 (0.29) | 1.40 (0.29) | 1.45 (0.29) |
| Model statistics | | | | | | |
| N | 184 | 184 | 184 | 184 | 184 | 184 |
| AIC | 1076.13 | 936.13 | 934.90 | 932.53 | 938.12 | 938.59 |
| ICC | 0.01 | 0.21 | 0.18 | 0.18 | 0.21 | 0.22 |

HRSV, human respiratory syncytial virus; Coef., coefficient of the variables; SE, standard error; AIC, Akaike's Information Criterion; ICC, intra-class correlation.

[a] P<0.05.

[b] P<0.01.

[c] P<0.001.

0.02 week respectively (Model 2: coefficient –0.18; P<0.01 and coefficient –0.02; P<0.01). Even after adjusting for meteorological variables and year variables, travelers' visits were independently associated with an earlier onset week of HRSV epidemic season. An increase in the number of inbound travelers from individual countries was associated with an earlier onset week of HRSV epidemic season such as Taiwan (Model 3: coefficient –0.07; P<0.01), South Korea (Model 4: coefficient –0.25; P<0.01), and China (Model 5: coefficient –0.09; P<0.05). On the other hand, an increase in the number of inbound travelers from Australia (Model 6: coefficient –0.60, P = 0.05) was marginally associated with an earlier onset week of the HRSV epidemic season. The onset week of HRSV epidemic was significantly earlier in 2017 and 2016 than in 2014 in all models. To this end, onset week of HRSV epidemic season was earlier in all models from 2014 to 2017.

## Discussion

In this study, we examined the associations between meteorological (mean temperature and relative humidity) conditions as well as the number of inbound overseas travelers, and the onset week of the HRSV epidemic season in Japan. Our findings suggest that these variables have a significant association with the onset week of HRSV epidemic season in Japan, and an increase of one unit in temperature, relative humidity and inbound travelers, are associated with 0.3, 0.2, 0.02 week earlier onset week in the year, respectively by multi-level analysis. To

our knowledge, this is the first study to evaluate the effects of inbound overseas travelers and meteorological conditions on the shift in HRSV epidemic season in Japan.

The HRSV epidemic period historically has been between September and December in Japan [18], however, recently the onset week of HRSV epidemic season has begun to shift, becoming earlier in the year since 2016 [4]. Studies in the past have focused on the epidemic peak and the size (e.g. number of HRSV cases) [24–26]. Although the number of HRSV cases is an important indicator, it is more useful to predict the onset week before the number of infections to increase. In a clinical setting, it is crucial to identify the start of an epidemic to allow paediatricians to assess whether or when to initiate palivizumab (anti-HRSV antibody) administration to prevent possible severe cases especially in high-risk children [27]. Our analysis suggests that not only meteorological conditions but also inbound overseas travelers may be associated with changing in the timing of HRSV epidemics, highlighting the need to consider these impacts in the timing of palivizumab administration. In April 2019, the Japan Pediatric Society published the consensus guidelines on the use of Palivizumab in Japan [28]. This guideline states that due to the shift of seasonality of HRSV in recent years in Japan, in order to increase the efficacy of palivizumab, it is necessary to raise the serum antibody titer to the level required for prophylaxis by the start of the onset week of HRSV epidemic season, and that pediatricians and others should lead a study in each prefecture to consider the month to start palivizumab and the duration and frequency of administration. In particular, they reported the shift of seasonality of HRSV in recent years in Japan, but they mentioned possible association with meteorological conditions as contributing factors. Therefore, our findings may bring a new information to the guideline that HRSV season may come earlier where they have more overseas travelers in the local area.

First, we examined the association between onset week of HRSV epidemic season and meteorological conditions. In previous studies, many researchers have reported on the association between meteorological conditions and HRSV transmission [3,5–11,13,24,29,30]. Most of the studies showed that higher temperature and humidity were associated with greater HRSV transmission in tropical and subtropical regions [3,6–9,11]. In contrast, in temperate zones, lower temperatures and higher humidity were associated with greater HRSV transmission [5]. In our results, we showed that higher temperature and/or higher humidity were associated with an earlier onset week of HRSV epidemic season. We previously reported that the number of HRSV cases in summer (from June-August) in Japan increased with higher temperature and higher relative humidity during 2007–2014 [26]. Our findings here also indicate that meteorological conditions have an association with HRSV transmission, corresponding with an early onset in the epidemic season. We observed a shift in the timing of the onset week of HRSV epidemic season, whereby, the epidemic shifted to occurring earlier in the year during summer. Global warming and consequences from climate change may affect the onset timing of HRSV epidemics. Meteorological condition in countries like Japan (i.e., temperate zones), which are near to the subtropical zone, might be affected by such subtle changes in climate.

Secondly, we considered the association between onset week of HRSV epidemic season and inbound overseas travelers to Japan. As the number of inbound overseas travelers has increased in recent years, there is a possibility that person-to-person close contact has increased, especially in densely populated cities such as urban areas (e.g., Tokyo, Osaka, Aichi, Fukuoka, and Hokkaido). An inflow of overseas travelers may affect the HRSV epidemic season owing to the infection route of HRSV having human-to-human transmission via droplets and contact with increased travelers resulting in greater contact rates [31]. In influenza, the travel has been reported to have an impact on the spread of infectious diseases [15,16], similarly, measles and novel coronavirus disease 2019 (COVID-19) have also been reported to be increased in transmission resulting from a greater traffic of people [13,14,17]. It is possible that

the inflow of travelers infected with HRSV may be associated with the HRSV epidemic season in Japan based on what was observed with the other infectious diseases. The results of this study suggest that the inflow of overseas travelers may have important impact on the outbreak of HRSV in Japan. However, there are few studies looking at HRSV transmission and the association with population behavioural and movement dynamics in Japan. In our study, after controlling for mean temperature, relative humidity, and year, we showed that the onset week of HRSV epidemic season was earlier with a greater number of inbound overseas travelers into Japan. Therefore, the onset week of HRSV epidemic season may have shifted due to the inflow of people from areas with different epidemic periods compared to that in Japan.

In this analysis, we expected that the impact on the onset week of HRSV epidemic season might be affected because of seasonality differences of HRSV in Taiwan and Australia compared to Japan. Epidemics of the HRSV often occur from June-September in Taiwan [29,32,33] and from June-September in temperate climate areas such as Sydney and Melbourne in Australia [8,11,34,35]. In contrast, HRSV epidemic tends occur from November-January in Korea and China as also typically seen before 2016 in Japan [30,36–39]. Thus, we expected that Taiwan and Australia would have a greater effect on causing an earlier onset week of HRSV epidemic season in Japan than Korea and China. However, all countries except Australia showed association with earlier onset of HRSV epidemic season in Japan. The marginal effect of Australia was presumably from the differences between the peak seasons for travelers to Japan (January and December) and that for HRSV in Australia (June-September). Further studies are needed to investigate the effect seasonality by the original location of where the travelers' have departed from and the timing of the HRSV epidemic season.

Several limitations of this study need to be considered. First, since this is an ecological study for limited years, the longitudinal monitoring and evaluation may be leading to a different result. Secondly, the travelers' data in this study does not take into account which local area of the country travelers have come from. The country areas are very large in China and Australia and have multiple climate zones. Therefore, the HRSV epidemic periods in each of these countries differ from area to area domestically, and the impact on the onset week of HRSV epidemic season may also differ. Thirdly, this study focused on onset week and did not analyse the peak week and size. Fourthly, this study does not consider the lag-time of inbound overseas travelers and meteorological conditions. The reason for this is that in the present analysis, we used 46 prefectures multiplied by 4 seasons (184 units), and used an annual bases for inbound overseas travelers and meteorological conditions instead of weekly bases. Therefore, a more detailed time-series level of analysis needs to be considered in our future studies. Finally, this study does not take into account on the numbers of outgoing Japanese travelers or changes in the domestic human mobility (e.g., inflow-outflow population between prefectures) over time. This may be a contributing factor in the changes of HRSV seasonality.

In conclusion, the earlier onset week of HRSV epidemic season in Japan is ecologically associated with an increased number of inbound overseas travelers, mean temperature, and relative humidity. We highlight the hypothesis that inbound overseas travelers may contribute to the shift in the HRSV epidemic season in addition to changes in meteorological conditions. The current findings provide useful information for public health decision making in prevention and the preparation for HRSV outbreaks in Japan. After the emergence of COVID-19, respiratory viral infections underwent significant changes. The number of HRSV cases has dramatically dropped and virtually no epidemics were observed except for small outbreaks in southernmost prefectures in Japan, such as Okinawa in November and Kagoshima in October in 2020 [19]. The dramatic decline of overseas travelers, strengthened personal hygiene behaviours and social infection control measurements may have contributed to such a decline of the HRSV cases in 2020 [40–45]. Thus it is important that future studies include post- 2020 data to

assess how the COVID-19 emergence changes the onset week of HRSV epidemic season. In particular, taking into account the impact of travel restrictions on the number of inbound overseas travelers, it may be possible to identify changes in the indirect impact on the onset week of HRSV epidemic season. Further evaluation and monitoring toward the change of respiratory infectious diseases after the COVID-19 pandemic should be jointly investigated at a global level by international coordinated schemes.

## Supporting information

**S1 Dataset. The weekly number of HRSV cases per sentinel site in Japan.**
(XLSX)

**S2 Dataset. The monthly number of overall inbound overseas travelers from all countries to Japan.**
(XLSX)

**S3 Dataset. The monthly number of inbound overseas travelers from Taiwan to Japan.**
(XLSX)

**S4 Dataset. The monthly number of inbound overseas travelers from South Korea to Japan.**
(XLSX)

**S5 Dataset. The monthly number of inbound overseas travelers from China to Japan.**
(XLSX)

**S6 Dataset. The monthly number of inbound overseas travelers from Australia to Japan.**
(XLSX)

**S7 Dataset. Dataset for the monthly mean temperature throughout Japan.**
(XLSX)

**S8 Dataset. Dataset for the monthly average relative humidity throughout Japan.**
(XLSX)

**S9 Dataset. Dataset of variables for Spearman's correlation analysis and multi-level analysis.**
(XLSX)

## Acknowledgments

We thank local governments, public health centers and institutes, and the National Institute of Infectious Diseases, Japan, for surveillance, laboratory testing, epidemiological investigations, and data collection. We thank Dr. Naohito Tanabe from the University of Niigata Prefecture in Japan for his useful comments. We would also like to thank Editage (www.editage.com) for English language editing.

## Author Contributions

**Conceptualization:** Keita Wagatsuma, Yugo Shobugawa, Reiko Saito.

**Data curation:** Keita Wagatsuma, Iain S. Koolhof.

**Formal analysis:** Keita Wagatsuma.

**Investigation:** Keita Wagatsuma, Iain S. Koolhof, Yugo Shobugawa, Reiko Saito.

**Project administration:** Reiko Saito.

**Writing – original draft:** Keita Wagatsuma.

**Writing – review & editing:** Iain S. Koolhof, Yugo Shobugawa, Reiko Saito.

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
