## [Decision Letter · Decision Letter 0]

25 Nov 2020

PONE-D-20-31262

Shifts in the epidemic season of human respiratory syncytial virus associated with inbound overseas travelers and meteorological conditions in Japan, 2014-2017: an ecological study

PLOS ONE

Dear Dr. Wagatsuma,

Thank you for submitting your manuscript to PLOS ONE. After careful consideration, we feel that it has merit but does not fully meet PLOS ONE’s publication criteria as it currently stands. Therefore, we invite you to submit a revised version of the manuscript that addresses the points raised during the review process.

We look forward to receiving your revised manuscript.

Kind regards,

Ngai Sze Wong

Academic Editor

PLOS ONE

2. Please include the date(s) on which you accessed the databases or records to obtain the data used in your study.

Reviewers' comments:

Reviewer's Responses to Questions

**Comments to the Author**

1. Is the manuscript technically sound, and do the data support the conclusions?

Reviewer #1: Yes

Reviewer #2: Partly

2. Has the statistical analysis been performed appropriately and rigorously? 

Reviewer #1: Yes

Reviewer #2: No

3. Have the authors made all data underlying the findings in their manuscript fully available?

Reviewer #1: Yes

Reviewer #2: Yes

4. Is the manuscript presented in an intelligible fashion and written in standard English?

Reviewer #1: Yes

Reviewer #2: Yes

5. Review Comments to the Author

Reviewer #1: Thank you for the opportunity to review this interesting manuscript

I just have some minor suggestions which I hope will help the authors improve their paper

1. Can the authors confirm the NIID data used came from the same sites during the study period (so that readers can be assured the authors are comparing the same target population across time)

2. Defining onset week = how is it possible to define the 101-st percentile rank? – by definition the maximum percentile rank is 100

3. Modelling with ‘onset week’ as an outcome is an interesting idea. It is not clear how temperature and travel numbers were included as variables in the model, but I suspect they were included cross-sectionally? Did the authors consider including a lag-time of 2-4 weeks in their models? We know from previous research that there is a lag of 2-4 weeks between temperature/travel changes and RSV detections. At the least the authors should conduct sensitivity analyses using different assumptions regarding lag-times

4. Could the meaning of the results be explained more clearly? E.g. “every 1 degree increase in temperature led to an earlier onset week by 0.18%” – I am not sure what 0.18% actually represents. It might help readers if this change could be reported in days or weeks (which is a unit that everyone understands)

5. Do authors have access to 2020 data? It would be very interesting to include data for this year to see how the onset week changes (given the effect of covid19 on travel visitors, but not on meteorological data)

6. Could the variables included in the model (travel numbers, temperature etc) be displayed in a Figure please? Similar to the current figure which shows how RSV cases change over calendar week. The current figure shows that 2017 has a distinct curve to the other years – it would be very interesting to see how the travel numbers/temperature etc for 2017 compare to other years.

Reviewer #2: The author’s work examining the associations between number of inbound travelers and meteorological factors on the onset week of HRSV epidemic season. Though an interesting and valuable subject, the methodologies and results were less robust and confusing. A few comments to improve the manuscript below:

Statistical analysis:

1. The authors use multivariate multiple linear regression to model the relationship between number of inbound travelers and meteorological factors on the onset week of HRSV epidemic season (page 13, lines 182-195). However, the regression model chosen by the authors seems to be less desirable:

a. Multivariate multiple linear regression is used to model the linear relationship between more than one dependent variable and more than one independent variables. However, in the present study, there was only ONE dependent variable (i.e. onset week of HRSV epidemic season by prefecture (N=46)) and the independent variables included: 1. Yearly mean temperature by prefecture; 2. Yearly relative humidity by prefecture; 3. Yearly number of inbound travelers (overall, Tai Wan, South Korea, China and Australia) by prefecture.

b. Furthermore, multivariate multiple linear regression is NOT suitable for longitudinal data with repeated measurements at multiple time point. In the present study, both independent and dependent variables were repeatedly measured in 2014, 2015, 2016 and 2017. Temporal autocorrelation might be present between data points, while the statistical methods used in this study could not account for it.

c. Revision of the statistical methods is recommended. In this case, multilevel model seems to be a more reasonable method for the present study, rather that rather than multivariate multiple linear regression. First level of the model account for the within prefecture’s time varying variation, while the second level account for the between prefecture variation.

2. Explain if the 4-years longitudinal data in the present study could provide sufficient power.

Results:

3. The results of the regression analysis were recommended to be reworked with a more appropriate statistical methods used so that it could provide more robust results.

Discussion:

4. While the onset week of HRSV epidemic season was shown to be earlier with a greater number of inbound overseas travelers. What are the specific public health implications of this study?

6. PLOS authors have the option to publish the peer review history of their article (what does this mean?). If published, this will include your full peer review and any attached files.

Reviewer #1: **Yes: **Robert Ware

Reviewer #2: No

---

## [Author Response · Author response to Decision Letter 0]

5 Mar 2021

Point by point response to the Editor and the Reviewers

We thank the editors and reviewers for their useful suggestions, which have helped us to improve the manuscript. The responses to all comments have been provided below.

- Editor

1. Please ensure that your manuscript meets PLOS ONE's style requirements, including those for file naming. The PLOS ONE style templates can be found at　https://journals.plos.org/plosone/s/file?id=wjVg/PLOSOne_formatting_sample_main_body.pdf and https://journals.plos.org/plosone/s/file?id=ba62/PLOSOne_formatting_sample_title_authors_affiliations.pdf

Response: Thank you for your important comments. We have checked that the manuscript has complied with the journal’s formatting guidelines and have corrected the reference formatting accordingly.

2. Please include the date(s) on which you accessed the databases or records to obtain the data used in your study.

Response: Thank you for your important suggestions. We have checked the access dates to the databases used in this study and made additional corrections to the References section (Pages 32-33, Lines 478-490) of the revised manuscript.

- Reviewer 1

1. Can the authors confirm the NIID data used came from the same sites during the study period (so that readers can be assured the authors are comparing the same target population across time).

Response: Thank you for your important comments. Unfortunately, we are unable to confirm that the National Institute of Infectious Diseases (NIID) under the Ministry of Health, Welfare and Labour in Japan (MHLW) data came from the same paediatric sentinel sites during the study period from 2014 to 2017. We are unable to resolve this issue as the problem is within the Japanese surveillance system of HRSV. Therefore, we have amended this subsection of the HRSV national surveillance data in Japan (Pages 7-8, Lines 88-113) to make these points clearer and added the following text.

Pages 8, Lines 98-105 “However, the information at the individual paediatric sentinel sites where each HRSV case count is reported is anonymised and not made public. Therefore, we could not specify whether these NIID data came from the same sentinel sites during the study period. Still we assume these changes of sentinel sites were minor and did not affect the quality of data. A sentinel site is located roughly over 30,000 populations according to the Infectious Disease Control Law in Japan and the prefecture government are responsible for maintaining the sites so that it does not affect the quality of data [21].”

2. Defining onset week = how is it possible to define the 101-st percentile rank? – by definition the maximum percentile rank is 100.

Response: Thank you for your important comments. We agree that it is important to clarify how the percentile of the epidemic onset week is defined. The definition of the onset week of the HRSV epidemic season is described in our paper, adapted from the method developed by Yamagami et al. The paper by Yamagami et al. does not mention in depth how the 101-st percentile ranks are defined, and we could not follow up on the reasons for these definitions. However, the validity of the determined onset week of HRSV season has been tested in their paper using data from 47 prefectures. Therefore, we have recognised the robustness of these estimates and have added the following to the Materials and methods section of the revised manuscript.

Pages 11, Lines 147-150 “The validity of the determined onset was confirmed by calculating the capture rate. The validity of the model was subsequently tested using data from 47 prefectures in the previous study [4].”

3. Modelling with ‘onset week’ as an outcome is an interesting idea. It is not clear how temperature and travel numbers were included as variables in the model, but I suspect they were included cross-sectionally? Did the authors consider including a lag-time of 2-4 weeks in their models? We know from previous research that there is a lag of 2-4 weeks between temperature/travel changes and RSV detections. At the least the authors should conduct sensitivity analyses using different assumptions regarding lag-times.

Response: Thank you for your important comments. We agree with you that the analysis should take into account a lag-time, which is an important factor in meteorological conditions contributing to number of inbound overseas travelers. However, in the present analysis, 46 prefectures multiplied by 4 seasons (N=184) were used as a unit, and meteorological conditions and the number of inbound overseas travelers were not on a weekly basis but on an annual basis. For this reason, this study does not include lag variables, and the analysis at a detailed level will be left for future work. Therefore, we have recognized the importance of these limitations and have added the following text to the Discussion section of the revised manuscript.

Pages 27-28, Lines 393-398 “Fourthly, this study does not consider the lag-time of inbound overseas travelers and meteorological conditions. The reason for this is that in the present analysis, we used 46 prefectures multiplied by 4 seasons (184 units), and used an annual bases for inbound overseas travelers and meteorological conditions instead of weekly bases. Therefore, a more detailed time-series level of analysis needs to be considered in our future studies.”

4. Could the meaning of the results be explained more clearly? E.g. “every 1 degree increase in temperature led to an earlier onset week by 0.18%” – I am not sure what 0.18% actually represents. It might help readers if this change could be reported in days or weeks (which is a unit that everyone understands).

Response: Thank you for your important comments. According to your suggestion, we have changed the wording to make it easier for the reader to understand. Specifically, we have amended the units of “%” to “week” in the revised manuscript as follows:

Pages 3, Lines 32-37 “Multi-level mixed-effects linear regression analysis showed that for every one person (per 100,000 population) increase in number of overall inbound overseas travelers led to an earlier onset week of HRSV epidemic season in the year by 0.02 weeks (coefficient –0.02; P<0.01). Higher mean temperature and higher relative humidity were also found to contribute to an earlier onset week by 0.30 weeks (coefficient –0.30; P<0.05) and 0.18 weeks (coefficient –0.18; P<0.01), respectively.”

Pages 19-20, Lines 279-294 “In Model 2, which included the number of overall inbound travelers, we found that every 1 degree increases in mean temperature led to an earlier onset week within the year by 0.30 week (Model 2: coefficient –0.30; P<0.05). In addition, every 1.0% increase in relative humidity and every 1 person (per 100,000 populations) increase in the number of overall inbound travelers were associated with an earlier onset week of the HRSV epidemic season by 0.18 week and 0.02 week respectively (Model 2: coefficient –0.18; P<0.01 and coefficient –0.02; P<0.01). Even after adjusting for meteorological variables and year variables, travelers’ visits were independently associated with an earlier onset week of HRSV epidemic season. An increase in the number of inbound travelers from individual countries was associated with an earlier onset week of HRSV epidemic season such as Taiwan (Model 3: coefficient –0.07; P<0.01), South Korea (Model 4: coefficient –0.25; P<0.01), and China (Model 5: coefficient –0.09; P<0.05). On the other hand, an increase in the number of inbound travelers from Australia (Model 6: coefficient –0.60, P=0.05) was marginally associated with an earlier onset week of the HRSV epidemic season. The onset week of HRSV epidemic was significantly earlier in 2017 and 2016 than in 2014 in all models. To this end, onset week of HRSV epidemic season was earlier in all models from 2014 to 2017.”

Pages 22-23, Lines 306-310 “Our findings suggest that these variables have a significant association with the onset week of HRSV epidemic season in Japan, and an increase of one unit in temperature, relative humidity and inbound travelers, are associated with 0.3, 0.2, 0.02 week earlier onset week in the year, respectively by multi-level analysis.”

5. Do authors have access to 2020 data? It would be very interesting to include data for this year to see how the onset week changes (given the effect of covid19 on travel visitors, but not on meteorological data).

Response: Thank you for your important comments. We believe that it would be very interesting to include and analyse the post- 2020 data to take into account the impact of coronavirus disease 2019 (COVID-19) in Japan. However, we were not able to include 2020 data in our evaluation as not all variables considered were published. Therefore, we have decided to leave this analysis for future work. In addition, we recognised the importance of considering these influences and have added the following text to the revised manuscript, which we have proposed as a topic for further research.

Pages 28-29, Lines 407-420 “After the emergence of COVID-19, respiratory viral infections underwent significant changes. The number of HRSV cases has dramatically dropped and virtually no epidemics were observed except for small outbreaks in southernmost prefectures in Japan, such as Okinawa in November and Kagoshima in October in 2020 [19]. The dramatic decline of overseas travelers, strengthened personal hygiene behaviours and social infection control measurements may have contributed to such a decline of the HRSV cases in 2020 [40-45]. Thus it is important that future studies include post- 2020 data to assess how the COVID-19 emergence changes the onset week of HRSV epidemic season. In particular, taking into account the impact of travel restrictions on the number of inbound overseas travelers, it may be possible to identify changes in the indirect impact on the onset week of HRSV epidemic season. Further evaluation and monitoring toward the change of respiratory infectious diseases after the COVID-19 pandemic should be jointly investigated at a global level by international coordinated schemes.” 

6. Could the variables included in the model (travel numbers, temperature etc) be displayed in a Figure please? Similar to the current figure which shows how RSV cases change over calendar week. The current figure shows that 2017 has a distinct curve to the other years – it would be very interesting to see how the travel numbers/temperature etc for 2017 compare to other years.

Response: Thank you for your important comments. As you requested, we have created additional figures showing the monthly changes in the number of inbound overseas travelers from Taiwan, South Korea, China, and Australia, mean temperature and relative humidity in Japan included in the model for study period from 2014 to 2017 (Fig 1). As a result of these changes, the following text descriptions and figure legends have been amended to the revised manuscript.

Pages 11-12, Lines 161-169 “The meteorological data published by the Japan Meteorological Agency (JMA) [22] was used as explanatory variables for the models in this paper. Monthly and yearly meteorological data observed at a meteorological observatory situated at the prefectural capital city was used for each respective prefecture. We extracted monthly and yearly mean temperature and relative humidity from 2014 to 2017. There were two meteorological observatories with missing data on relative humidity. Therefore, we selected the observatories sites with the nearest distance from the prefectural capital by substitution. Using these monthly meteorological data in each prefecture, mean temperature and relative humidity over the whole of Japan have been calculated.”

Pages 12, Lines 172-178 “As a surrogate of the monthly and yearly number of overall inbound overseas travelers by prefecture, the daily number of foreign visitors accommodated (e.g., hotels) in each prefecture from 2014 to 2017 were sourced from the travel statistics survey of the Japan National Tourism Organization (JNTO) [23]. In addition, we used yearly number of travelers by country from the top three countries (China, South Korea, and Taiwan) with the most travelers coming into Japan. In addition, Australia was used as the country with the largest number of travelers from the southern hemisphere from 2014 to 2017.”

Pages 15-16, Lines 224-246 “Seasonality trends of HRSV cases, mean temperature, relative humidity, and inbound overseas travelers throughout Japan, 2014-2017 HRSV activity in Japan usually peaks in the winter season (from December-January). The peak epidemic of HRSV was observed in December in 2014 and 2015, but was changed to October in 2016, and to September in 2017 (Fig 1A, S1 Dataset). Correspondingly, the annual onset week of the HRSV epidemic season occurred earlier in the later year. The median onset weeks of the HRSV epidemic season were around 36.5 weeks (September) in 2014, 36.0 weeks (September) in 2015, 35 weeks (late August) in 2016, and 30 weeks (beginning of July) in 2017, respectively. The median onset week of HRSV epidemic season was approximately 6.0 weeks (1.5 months) earlier in 2017 than in 2014. On the other hand, the number of inbound overseas travelers increased year on year, doubled from 13,413,467 in 2014 to 28,691,073 in 2017 (S2 Dataset). However, the monthly number of inbound overseas travelers from all countries during 2014 and 2017 did not show clear seasonal trend (Fig 1B and S2 Dataset). The number of inbound overseas travelers from Taiwan, China and South Korea remained almost steady through the year, while Australia is observed to have a higher number of inbound travelers in the winter months of January and December (Fig 1C to 1F, and S3-S6 Dataset). This pattern did not differ during the four years. The mean temperature peaks at approximately 27 ℃ in August of each year and the relative humidity is around 60-70% throughout the year in Japan (Fig 1G and H, S7 and S8 Dataset). Mean temperature and relative humidity did not vary significantly throughout the 4-year period.”

Pages 17, Lines 248-255 “Fig 1. Trends of HRSV cases, mean temperature, relative humidity, and inbound overseas travelers throughout Japan, 2014-2017 (A) Epidemic curve of HRSV in Japan from 2014 to 2017. (B) Monthly number of overall inbound overseas travelers from all countries during 2014 and 2017, (C) Taiwan, (D) South Korea, (E) China, (F) Australia, (G) Monthly mean temperature (℃) throughout Japan from 2014 to 2017, (H) Monthly average relative humidity (%) from 2014 to 2017. The orange, yellow, green and brown lines represent 2014, 2015, 2016 and 2017, respectively.”

- Reviewer 2

Comments:

1. The authors use multivariate multiple linear regression to model the relationship between number of inbound travelers and meteorological factors on the onset week of HRSV epidemic season (page 13, lines 182-195). However, the regression model chosen by the authors seems to be less desirable: a. Multivariate multiple linear regression is used to model the linear relationship between more than one dependent variable and more than one independent variables. However, in the present study, there was only ONE dependent variable (i.e. onset week of HRSV epidemic season by prefecture (N=46)) and the independent variables included: 1. Yearly mean temperature by prefecture; 2. Yearly relative humidity by prefecture; 3. Yearly number of inbound travelers (overall, Tai Wan, South Korea, China and Australia) by prefecture. b. Furthermore, multivariate multiple linear regression is NOT suitable for longitudinal data with repeated measurements at multiple time point. In the present study, both independent and dependent variables were repeatedly measured in 2014, 2015, 2016 and 2017. Temporal autocorrelation might be present between data points, while the statistical methods used in this study could not account for it. c. Revision of the statistical methods is recommended. In this case, multilevel model seems to be a more reasonable method for the present study, rather that rather than multivariate multiple linear regression. First level of the model account for the within prefecture’s time varying variation, while the second level account for the between prefecture variation.

Response: Thank you for your important comments. As you pointed out, we have applied a multi-level analysis more appropriate to this longitudinal data. Specifically, we conceptualised our analysis with a multi-level structure consisting of factors nested within prefectures to account the temporal variations between 2014 and 2017 (Table 2). Then, we evaluated how mean temperature, relative humidity, and the number of inbound overseas travelers affect the early onset week of HRSV epidemic season. Therefore, we have made substantial amendments to the Materials and methods and Results sections of the revised manuscript and added the following text.

Pages 14-15, Lines 198-2189 “Secondly, multi-level mixed-effects linear regression model was performed to adjust for the effects of confounding variables (N=4×46=184). We conceptualised our analysis with a multi-level structure consisting of factors nested within prefectures (prefectural-level) to account for the impact of year repeated measures. We fitted the data using a multi-level linear regression procedure with a mixed effect model, adjusting for both individual and prefectural levels as fixed effects. The parameters of the model were chosen to be random effects estimators using the maximum likelihood. The onset week of HRSV epidemic season by each year (continuous) was the dependent variable, and yearly number of inbound travelers (continuous) and meteorological (yearly mean temperature and relative humidity) conditions (continuous) were included as explanatory variables. Furthermore, the model was adjusted by inputting year variables (years 2014, 2015, 2016, and 2017) (dummy variables) as a covariate to control seasonal variation. Our first analysis involved the estimation of a null model (Model 1). The null model allows us to decompose the variance in the onset week of HRSV epidemic season to determine whether it is due to variation at the prefectural or individual level. Model 2 represents the number of overall inbound travelers per 100,000 population. Models 3, 4, 5, and 6 consider the numbers of inbound travelers per 100,000 population from Taiwan, South Korea, China, and Australia, respectively. The regression coefficients and P-values were calculated in each multi-level analysis. Goodness-of-fit statistics of each model was evaluated by calculating the Akaike's Information Criterion values (AIC). The intra-class correlation (ICC) was calculated to evaluate the similarity within group.”

Pages 19-20, Lines 275-294 “The multi-level analysis showed that mean temperature, relative humidity, and the number of inbound travelers were negatively associated with the onset week of the HRSV epidemic season (Table 2, S9 Dataset). The null model with no predictors (Model 1) revealed no significant evidence of inter-prefectural variation (prefectural-level variance=0.61, ICC=0.01). In Model 2, which included the number of overall inbound travelers, we found that every 1 degree increases in mean temperature led to an earlier onset week within the year by 0.30 week (Model 2: coefficient –0.30; P<0.05). In addition, every 1.0% increase in relative humidity and every 1 person (per 100,000 populations) increase in the number of overall inbound travelers were associated with an earlier onset week of the HRSV epidemic season by 0.18 week and 0.02 week respectively (Model 2: coefficient –0.18; P<0.01 and coefficient –0.02; P<0.01). Even after adjusting for meteorological variables and year variables, travelers’ visits were independently associated with an earlier onset week of HRSV epidemic season. An increase in the number of inbound travelers from individual countries was associated with an earlier onset week of HRSV epidemic season such as Taiwan (Model 3: coefficient –0.07; P<0.01), South Korea (Model 4: coefficient –0.25; P<0.01), and China (Model 5: coefficient –0.09; P<0.05). On the other hand, an increase in the number of inbound travelers from Australia (Model 6: coefficient –0.60, P=0.05) was marginally associated with an earlier onset week of the HRSV epidemic season. The onset week of HRSV epidemic was significantly earlier in 2017 and 2016 than in 2014 in all models. To this end, onset week of HRSV epidemic season was earlier in all models from 2014 to 2017.”

2. Explain if the 4-years longitudinal data in the present study could provide sufficient power.

Response: Thank you for your important comments. In multi-level analysis, we considered the year variables (years 2014, 2015, 2016 and 2017) as explanatory variables and examined the significant variation over the four years. Therefore, we have added the following text to the results section of the revised manuscript.

Pages 20, Lines 292-294 “The onset week of HRSV epidemic was significantly earlier in 2017 and 2016 than in 2014 in all models. To this end, onset week of HRSV epidemic season was earlier in all models from 2014 to 2017.”

3. The results of the regression analysis were recommended to be reworked with a more appropriate statistical methods used so that it could provide more robust results.

Response: Thank you for your important comments. As you pointed out, we have changed our statistical analysis to be more robust by performing a multi-level analysis, more appropriate to the structure of the data. Specifically, we set 46 prefectures as the second level (prefectural-level) and analysed data obtained four times longitudinally in the same prefecture, taking into account differences in trends between prefectures (onset week HRSV epidemic season are starting earlier or later). Then, we took into account meteorological conditions and the number of inbound overseas travelers to allow for a more valid analysis. Therefore, we have made the following changes to the statistical analysis subsection of the Materials and methods section of the revised manuscript.

Pages 14-15, Lines 198-2189 “Secondly, multi-level mixed-effects linear regression model was performed to adjust for the effects of confounding variables (N=4×46=184). We conceptualised our analysis with a multi-level structure consisting of factors nested within prefectures (prefectural-level) to account for the impact of year repeated measures. We fitted the data using a multi-level linear regression procedure with a mixed effect model, adjusting for both individual and prefectural levels as fixed effects. The parameters of the model were chosen to be random effects estimators using the maximum likelihood. The onset week of HRSV epidemic season by each year (continuous) was the dependent variable, and yearly number of inbound travelers (continuous) and meteorological (yearly mean temperature and relative humidity) conditions (continuous) were included as explanatory variables. Furthermore, the model was adjusted by inputting year variables (years 2014, 2015, 2016, and 2017) (dummy variables) as a covariate to control seasonal variation. Our first analysis involved the estimation of a null model (Model 1). The null model allows us to decompose the variance in the onset week of HRSV epidemic season to determine whether it is due to variation at the prefectural or individual level. Model 2 represents the number of overall inbound travelers per 100,000 population. Models 3, 4, 5, and 6 consider the numbers of inbound travelers per 100,000 population from Taiwan, South Korea, China, and Australia, respectively. The regression coefficients and P-values were calculated in each multi-level analysis. Goodness-of-fit statistics of each model was evaluated by calculating the Akaike's Information Criterion values (AIC). The intra-class correlation (ICC) was calculated to evaluate the similarity within group.”

4. While the onset week of HRSV epidemic season was shown to be earlier with a greater number of inbound overseas travelers. What are the specific public health implications of this study?

Response: Thank you for your important comments. In this study, we highlighted the earlier onset week of HRSV epidemic season may be associated with an increase in the number of inbound overseas travelers in Japan, after adjusting meteorological conditions (mean temperature and relative humidity). Therefore, our findings may bring new information to inform guidelines that the HRSV season may come earlier in areas which have greater overseas travelers. Therefore, we have added the following text to the Discussion section of the revised manuscript in recognition of the public health utility of this study.

Pages 23-24, Lines 321-334 “Our analysis suggests that not only meteorological conditions but also inbound overseas travelers may be associated with changing in the timing of HRSV epidemics, highlighting the need to consider these impacts in the timing of palivizumab administration. In April 2019, the Japan Pediatric Society published the consensus guidelines on the use of Palivizumab in Japan [28]. This guideline states that due to the shift of seasonality of HRSV in recent years in Japan, in order to increase the efficacy of palivizumab, it is necessary to raise the serum antibody titer to the level required for prophylaxis by the start of the onset week of HRSV epidemic season, and that pediatricians and others should lead a study in each prefecture to consider the month to start palivizumab and the duration and frequency of administration. In particular, they reported the shift of seasonality of HRSV in recent years in Japan, but they mentioned possible association with meteorological conditions as contributing factors. Therefore, our findings may bring a new information to the guideline that HRSV season may come earlier where they have more overseas travelers in the local area.”

- Other amendments made by the authors

These sections were amended by the authors for clarification (shown in yellow highlights in the revised manuscript).

Pages 4, Lines 42-44 “The impact of international travelers on seasonality of HRSV can be further extended to investigations on the changes of various respiratory infectious diseases especially after the coronavirus disease 2019 (COVID-19) pandemic.” 

Pages 5, Lines 59-60 “Until the start of coronavirus disease 2019 (COVID-19) pandemic in 2020,”

Pages 6, Lines 64-66 “Indeed, we recently reported that early cases of COVID-19 in January-February 2020 in Japan were concentrated in prefectures where the numbers of foreign travelers were high [17].” 

Pages 6, Lines 67-71 “This study aims to test whether the number of inbound overseas travelers and meteorological conditions are associated with the onset week of HRSV epidemic season in Japan. We included the number of inbound overseas travelers as a parameter as well as meteorological conditions, to examine the associations with the onset week of HRSV epidemic season during 2014-2017.”

Pages 18, Lines 264-265 “These results suggested the increased temperature and the higher number of overseas travelers are associated to earlier onset of HRSV season, respectively.”

Pages 22, Lines 304-306 “In this study, we examined the associations between meteorological (mean temperature and relative humidity) conditions as well as the number of inbound overseas travelers, and the onset week of the HRSV epidemic season in Japan.”

Pages 27, Lines 379-383 “However, all countries except Australia showed association with earlier onset of HRSV epidemic season in Japan. The marginal effect of Australia was presumably from the differences between the peak seasons for travelers to Japan (January and December) and that for HRSV in Australia (June-September).”

Pages 27, Lines 386-388 “First, since this is an ecological study for limited years, the longitudinal monitoring and evaluation may be leading to a different result.”

The following references have been added to the References section as part of the revision (Pages 33, Lines 484-486; Pages 34, Lines 500-505; Pages 36-37, Lines 539-559).

・21. National Institute of Infectious Diseases (NIID). National Epidemiological Surveillance for Infectious Diseases (NESID) in Japan. Available from: https://www.niid.go.jp/niid/images/epi/nesid/nesid_ja.pdf. (accessed Mar 1, 2021).

・27. The IMpact-RSV Study Group. Palivizumab, a humanized respiratory syncytial virus monoclonal antibody, reduces hospitalization from respiratory syncytial virus infection in high-risk infants. Pediatrics. 1998;102:531-7. pmid: 9738173.

・28. Japan Pediatric Society. Consensus guidelines for the use of palivizumab in Japan. Available from: http://www.jpeds.or.jp/uploads/files/20190402palivizumabGL.pdf. (accessed Jan 8, 2021).

・40. Britton PN, Hu N, Saravanos G, Shrapnel J, Davis J, Snelling T, et al. COVID-19 public health measures and respiratory syncytial virus. The Lancet Child & Adolescent Health. 2020;4(11):e42-e3. pmid:32956616.

・41. Baker RE, Park SW, Yang W, Vecchi GA, Metcalf CJE, Grenfell BT. The impact of COVID-19 nonpharmaceutical interventions on the future dynamics of endemic infections. Proc Natl Acad Sci U S A. 2020;117(48):30547-53. pmid:33168723.

・42. Nolen LD, Seeman S, Bruden D, Klejka J, Desnoyers C, Tiesinga J, et al. Impact of Social Distancing and Travel Restrictions on non-COVID-19 Respiratory Hospital Admissions in Young Children in Rural Alaska. Clin Infect Dis. 2020 Sep 5;ciaa1328. pmid:32888007.

・43. Yeoh DK, Foley DA, Minney-Smith CA, Martin AC, Mace AO, Sikazwe CT, et al. The impact of COVID-19 public health measures on detections of influenza and respiratory syncytial virus in children during the 2020 Australian winter. Clin Infect Dis. 2020 Sep 28;ciaa1475. pmid:32986804.

・44. Trenholme A, Webb R, Lawrence S, Arrol S, Taylor S, Ameratunga S, et al. COVID-19 and Infant Hospitalizations for Seasonal Respiratory Virus Infections, New Zealand, 2020. Emerg Infect Dis. 2021;27(2). pmid:33263515.

・45. Kuitunen I, Artama M, Makela L, Backman K, Heiskanen-Kosma T, Renko M. Effect of Social Distancing Due to the COVID-19 Pandemic on the Incidence of Viral Respiratory Tract Infections in Children in Finland During Early 2020. Pediatr Infect Dis J. 2020;39(12):e423-e7. pmid:32773660.

To clarify the results in Fig 1 and Tables 1 and 2 of the main analysis, we have created data tables for all supplements used in these analyses and added them to the Supporting information section (Pages 37-38, Lines 562-573).

Other small typographical errors were corrected and words were added for clarification in the revised manuscript.

---

## [Decision Letter · Decision Letter 1]

9 Mar 2021

Shifts in the epidemic season of human respiratory syncytial virus associated with inbound overseas travelers and meteorological conditions in Japan, 2014-2017: an ecological study

PONE-D-20-31262R1

Dear Dr. Wagatsuma,

We’re pleased to inform you that your manuscript has been judged scientifically suitable for publication and will be formally accepted for publication once it meets all outstanding technical requirements.

Kind regards,

Ngai Sze Wong

Academic Editor

PLOS ONE

Reviewer's Responses to Questions

**Comments to the Author**

1. If the authors have adequately addressed your comments raised in a previous round of review and you feel that this manuscript is now acceptable for publication, you may indicate that here to bypass the “Comments to the Author” section, enter your conflict of interest statement in the “Confidential to Editor” section, and submit your "Accept" recommendation.

Reviewer #1: All comments have been addressed

Reviewer #2: All comments have been addressed

2. Is the manuscript technically sound, and do the data support the conclusions?

Reviewer #1: Yes

Reviewer #2: Yes

3. Has the statistical analysis been performed appropriately and rigorously? 

Reviewer #1: Yes

Reviewer #2: Yes

4. Have the authors made all data underlying the findings in their manuscript fully available?

Reviewer #1: No

Reviewer #2: Yes

5. Is the manuscript presented in an intelligible fashion and written in standard English?

Reviewer #1: Yes

Reviewer #2: Yes

6. Review Comments to the Author

Reviewer #1: Thank you for the thorough response to the Reviewer comments.

Thank you for the thorough response to the Reviewer comments.

Reviewer #2: All the comments has been addressed. The revised version of the manuscript seems to be much more easy to follow, with appropriate statistical methods applied and more robust results provided.

7. PLOS authors have the option to publish the peer review history of their article (what does this mean?). If published, this will include your full peer review and any attached files.

Reviewer #1: **Yes: **Robert S Ware

Reviewer #2: No

---

## [Editor Report · Acceptance letter]

12 Mar 2021

PONE-D-20-31262R1 

Shifts in the epidemic season of human respiratory syncytial virus associated with inbound overseas travelers and meteorological conditions in Japan, 2014-2017: an ecological study 

Dear Dr. Wagatsuma:

I'm pleased to inform you that your manuscript has been deemed suitable for publication in PLOS ONE. Congratulations! Your manuscript is now with our production department. 

Kind regards, 

on behalf of

Dr. Ngai Sze Wong 

Academic Editor

PLOS ONE